# Technical and Economic Analysis of the Implementation of Selected Variants of Road Investment

**Marcin Szczepański \*** and **Beata Grzyl**

Faculty of Civil and Environmental Engineering, Gdansk University of Technology, 80-233 Gdansk, Poland; beata.grzyl@pg.edu.pl
**\*** Correspondence: marszcze@pg.edu.pl

**Abstract:** The aim of the article is to analyze three variants of modernization and reconstruction of a road intersection, which in practice is the cause of numerous collisions and accidents. Detailed design solutions are presented for them. The aim of the analyses is to indicate an effective solution that, taking into account technical modifications of the road system elements, will ensure the functionality of the road system to the highest degree and significantly reduce the number of road incidents. To indicate the optimal solution, quantitative data (cost and duration of activities for three options) and qualitative data (determined based on the own experience and knowledge of road industry experts) is analyzed. The authors refer to many criteria of various natures (e.g., economic, technical, functional, environmental, social), which allow for comprehensive consideration of the current requirements of road users and changing circumstances, among others a steady increase in the number of vehicles and growing social expectations in terms of road parameters. Considering the presented analyses and arguments, the authors recommend option 1 as optimal. This is the most expensive solution among those analyzed and with the longest implementation time, however, taking into account the long-term prognosis of the direction and scope of changes to the existing standards and requirements for road infrastructure, it can be stated that option 1 meets them to the highest degree, and also has the greatest potential. The envisaged solution ensures high standards of the quality of road infrastructure use in terms of functionality, capacity, technical parameters, as well as the safety of traffic participants related to the smoothness of the journey, reduction of the number of collisions and accidents.

**Keywords:** economic analysis; road investment; optimization; cost estimates; roads; investment variants

---

## 1. Introduction

The significant economic development observed in many countries around the world over the last few decades has caused many positive social changes, e.g., increasing the number of inhabitants, and improving the quality of life. In practice, there is a growing demand for road transport and, at the same time, ever-increasing requirements of road users regarding the quality of travel and an increase in the number of vehicles traveling on the roads, and as a result an increase of traffic volume and sensitivity of road systems to interference and rapid degradation of infrastructure [1]. For many years, road transport has been recognized as one of the sectors that is important for the economic development of countries, and at the same time the least safe (due to the highest accident rate) and characterized by a very large adverse impact on the environment. It causes severe air pollution and noise emissions, which affects the health and quality of life of people who are within the range of roads [2]. The task of road infrastructure is to meet the need for safe and economic transport of people and goods [3]. Therefore, it must meet certain standards of quality of use (including functionality, capacity, and technical parameters) and safety of traffic participants (including smooth travel, limiting the number of collisions and accidents and negative impact on

the environment) [4]. The above standards must be considered already at the stage of infrastructure design, but also maintained throughout the period of its functioning [5–7]. The owners and managers of road infrastructure, represented by governmental institutions, are responsible for ensuring proper road traffic conditions on managed road networks and for ongoing compliance with increasing social requirements regarding a high level of service, smooth road travel and low environmental impact of road transport on the local environment. Due to the often very limited financial resources at their disposal, in practice they face the problem of effective spending [8]. Their decision, in many cases, is to find a compromise between maintaining the current standard of the existing road network and implementing a new solution, considering the growing needs and future requirements of users.

The authors analyze three cases of investment implementation from the point of view of the entity responsible for managing a given road section, considering specific criteria and real possibilities as well as conditions in the scope of project implementation. Based on the analyses, the authors indicate a rational solution for the reconstruction and modernization of the collision intersection. This solution considers current and future transport needs, road users' expectations, social requirements and meets sustainable infrastructure development in terms of reliability, user safety and limited environmental impact.

It should be emphasized that the analyses presented in the article should be treated as preliminary studies of the diagnosed problem. At this stage, quantitative (cost and time) and qualitative (based on expert knowledge) data are analyzed. The authors are aware that the optimization of issues related to construction investments, which are characterized by high specificity and an interdisciplinary character, is particularly complicated, and there are numerous restrictions on the use of methods in the process of conducting it. According to the authors' assumption, work on the issue will be continued.

## 2. Optimization Issues in Source Literature

Optimization aims to indicate the best possible solution, selected from a specific set, based on the adopted criteria, which are expressed using mathematical functions, i.e., objective functions. In practice, one-criterion optimization is widely used. Its goal is to find the optimal solution considering one criterion (e.g., minimizing the time of project implementation or maximizing the company's profit). It should be emphasized that global optima (absolutely the best solutions), but also local optima (the best solutions in a certain environment) can occur for the examined objective function. For this reason, in practice, at the stage of results interpretation and decision making, multi-criteria optimization brings better results [9]. Multi-criteria decision problems can be classified into the following groups [10,11]:

- Multiple-attribute decision problem (MADP), for which there is a limited number of decision options, each of which has a specific, pre-determined and associated level of achievement of the features considered by the decision-maker to be significant, and the decision is made on their basis.
- Multiple-objective decision problem (MODP) that do not have a predetermined number of options with values of features specific to the problem; however, these problems have a set of quantifiable goals on the basis of which a decision is made and a set of well-defined restrictions on the value of decision variables of possible options.

In the practice of the activities of construction companies, the main elements of optimization, and at the same time the areas of investment project management are project implementation time, budget foreseen for it and resources assigned to it. Searching for an optimal solution that considers two main criteria is a common activity: minimizing the time and costs of implementing the project. In this case, it is important that the shortest duration of a construction investment is usually not the same as the lowest cost. On the total cost curve (kc), Figure 1, one can indicate, among others two values important from the point of view of optimization [12]:

- $t_n$, understood as the normal time of implementation of the project, which corresponds to the lowest cost of its implementation ($cu_{min}$);

- tgr understood as the cut-off time, i.e., the shortest possible duration of the project to which the cut-off cost (kgr) corresponds.

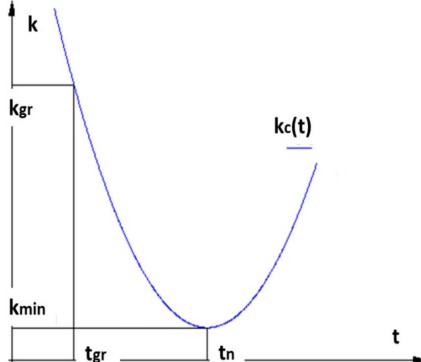

**Figure 1.** Dependence of the costs of the undertaking on the time of its implementation [12].

The analysis of different variants of the execution of works and the corresponding times and costs allows to determine the total cost curve. If the duration of the project exceeds normal time, an increase in the investment cost is observed. This means that the optimal sought solution is between the cut-off time and normal time. The choice of the best solution depends in this case on the preferences of the decision maker. Optimization methods (including, among others, schedules) can be indicated [13,14]:

- accurate;
- mathematical (e.g., linear programming, dynamic programming, branch and bound method);
- heuristics (e.g., priority heuristics);
- metaheuristic (e.g., genetic algorithms).

However, it should be emphasized that the optimization of issues occurring in practice, in the field of construction, is particularly complicated. This means that the time needed to solve the problems increases exponentially as the size of investment projects increases [15–17]. There are also numerous restrictions on the use of accurate and heuristic methods to find the optimal solution. These methods do not allow, among others, a solution to be obtained to the complex problems of scheduling construction works in an acceptable time [12]. There are, among others a serious problem with scheduling tasks, which is caused by the interdisciplinary nature of construction projects (technical, economic, logistical, legal issues, etc.) and their large specificity (unique location, different scale of investment, different availability of resources, sometimes unique technology) [18–21]. Metaheuristic algorithms are a valuable tool for solving complex problems in the field of construction and scheduling (task scheduling) [22–24]. However, they do not guarantee finding the best possible solution. At the same time, it is obvious that the results obtained by using any of the optimization methods are strictly dependent on the given input parameters.

## 3. Purpose of the Article

The aim of the article is to analyze the variants of modernization and reconstruction of a selected road intersection and on this basis to indicate the optimal solution. The source material is the documentation made available by the contracting authority [25]. The article presents three selected variants (out of six subjects to initial qualification) for which detailed design solutions have been proposed. Following the SWOT analysis (Strengths, Weaknesses, Opportunities, Threats), three suggestions for solutions were eliminated from further research. It is assumed that the result of the analysis is to indicate a solution that takes into account the modification of technical elements of the road system, which will ensure the functionality of the road system in question and will reduce the number of collisions and accidents. In order to indicate the optimal solution, quantitative data

(cost and time of implementation of activities for each of the three selected variants) and qualitative (determined based on own experience and knowledge of road industry experts) is analyzed [26–28]. During the analysis, aimed at indicating a rational and effective solution of the road system in question, many criteria of various natures (e.g., economic, technical, functional, environmental, social) are used. This systemic approach makes it possible to take into account the current requirements of road users, but also the circumstances that change over time, i.e., a steady increase in the number of vehicles and growing social expectations in terms of road parameters.

## 4. Description of the Existing State-Research Problem

In its present condition, the existing main road is connected by a collision junction with a subordinate-commune road (see Figures 2 and 3). The condition of asphalt pavement on the section of the main road covered by the study, the authors assessed (based on the local inspection) as quite good.

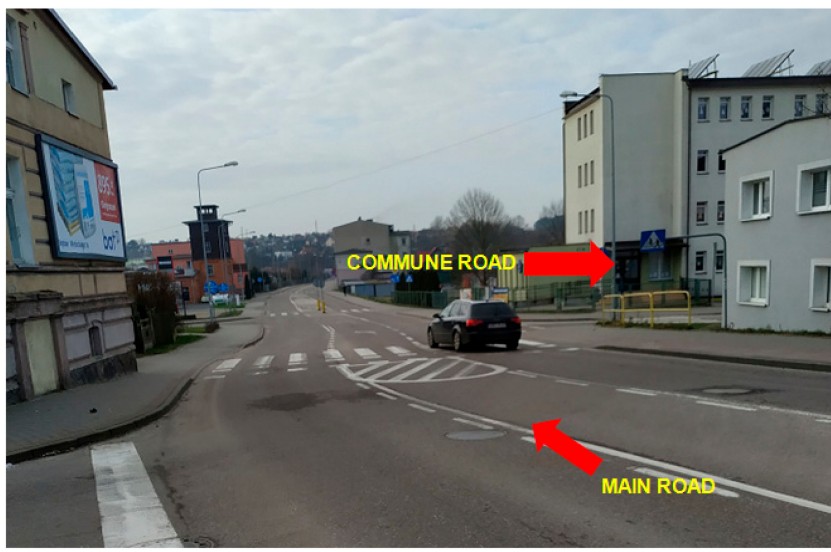

**Figure 2.** View of the intersection of the main road with the commune road.

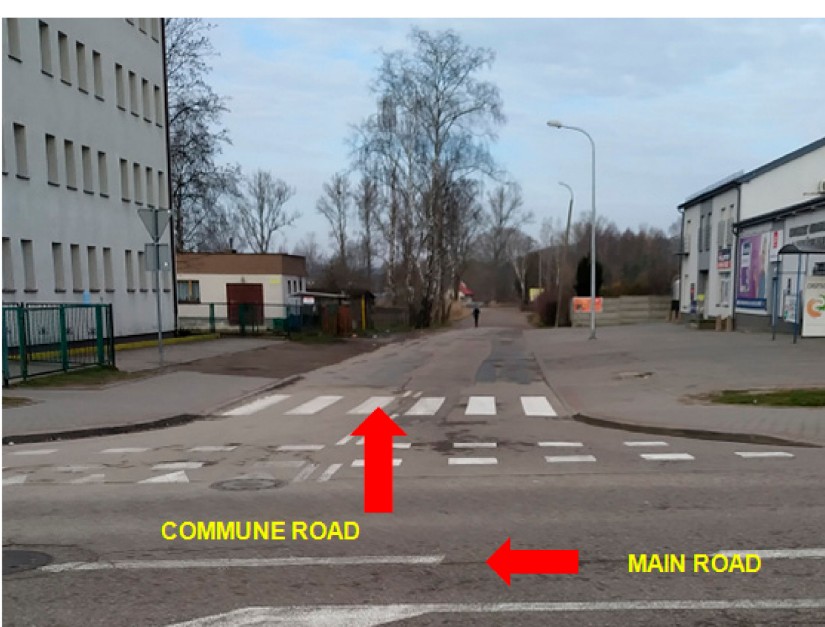

**Figure 3.** The current solution of connecting the main road with the commune road.

In the initial part of the section, the commune road has an asphalt surface, while in the further part it is a dirt road. On the asphalt and dirt section of the commune road, numerous cracks, ruts, and damage were diagnosed because of its continuous operation. The construction of the commune road is not adapted to the conditions in the area, which means that the road surface along its entire section requires deep reconstruction. The condition of the commune road was assessed as poor.

Within the planned investment there are downtown buildings-numerous residential and service buildings, a kindergarten, a gas station, and small shops. On both sides of the main road there are pedestrian and bicycle paths, followed by concrete pavement walkways (Figure 4).

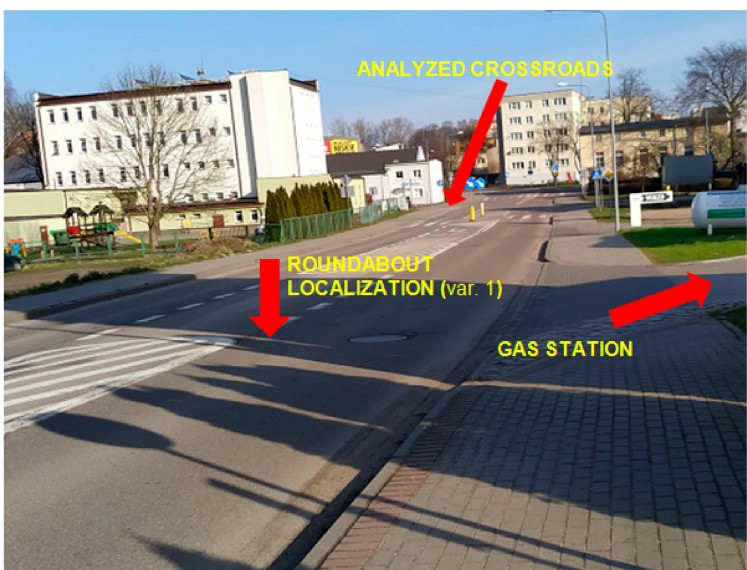

**Figure 4.** Destination location of the roundabout in option 1.

The pedestrian-bicycle route crossing the commune pedestrian road from the inrun side is fenced in sections with safety barriers and marked with vertical markings. Poor traffic organization and insufficient security are major problems. The road exit from a commune road to a main road does not provide sufficient visibility. From the left, visibility is significantly limited by the service building, and from the right by the kindergarten building. For this reason, the trip is very collision-prone and poses a great threat to users of traffic, both pedestrian and wheeled. In addition, there is a pedestrian crossing near the intersection, which further reduces safety and visibility. Another element posing a threat to traffic users (including children) is the dangerous location of the exit from the kindergarten-in the immediate vicinity of the commune road. The lack of good, double-sided visibility at the exit of the commune road, pedestrian traffic and oncoming bikes mean that drivers are not able to properly assess the situation at the intersection.

Numerous utilities in the form of gas, electricity, teletechnical and sewage networks are located within the road intersection and in its immediate vicinity the area is densely equipped. There are low green areas and local trees near the pavements and bicycle routes. The land was defined as low bearing, assigned to category G3 [29].

At the intersection, which is the subject of the analysis, there have been numerous road incidents and collisions in recent years caused by improper traffic organization. The use of the intersection in its current form (Figure 2) generates numerous traffic problems for the contracting party, primarily such as: low throughput, long waiting time for travel from subordinate roads, failure to ensure an adequate level of safety of road participants (drivers, passengers and pedestrians). An important parameter associated with the use of the intersection in its current form is also a significant limitation of visibility, which has a decisive impact on the ability to properly assess the traffic situation at the intersection, make a decision and execute the intended maneuver by road users-drivers and pedestrians at crossings.

As a result, the current solution at the intersection is the cause of numerous traffic incidents. It should also be emphasized that an important aspect of future road works is their proper protection, which also has a significant impact on safety [30,31].

The current-insufficient technical parameters and significantly limited capacity in terms of throughput and security have prompted the contracting authority to seek a solution to the problem, i.e., to introduce changes in the existing intersection system, taking into account the possibility of occupying an additional area.

*4.1. Three Suggestions for Solving the Problem—Variants That Are the Subject of Research*

To solve the problem faced by the contracting authority, three variants of reconstruction and modernization of the intersection are considered:

1.  construction of a new roundabout on a main road at 30 m from the original intersection, partly on private and investor-owned land (option I)—see Figure 5;
2.  construction of a new roundabout on a main road at the place of the existing intersection, partly on private and investor-owned land (option II);
3.  construction of a new roundabout on a main road at the place of the existing intersection, partly on private and investor-owned land (option III).

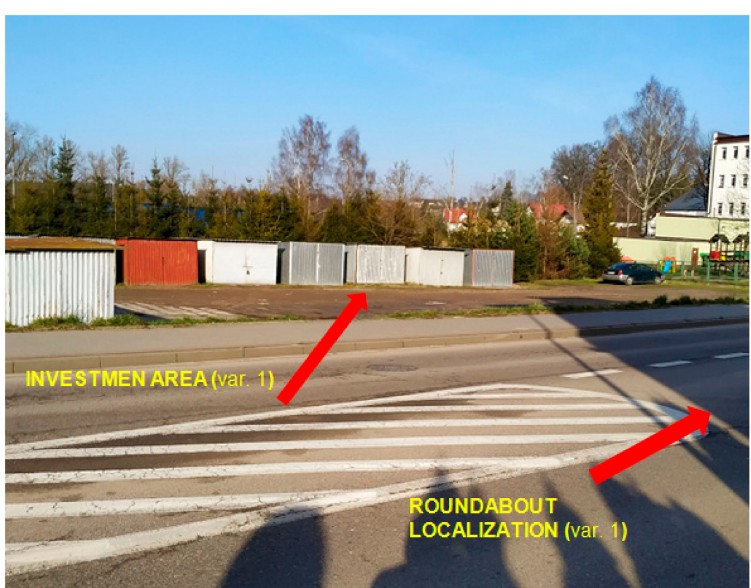

**Figure 5.** Location of the roundabout, land taken over for the investment in option No. 1.

For the purposes of this article, project documentation was prepared for the three proposed variants, and on that basis descriptions of the designed states were prepared and the analyses are presented below. It is obvious that, depending on the type of solution used, the reconstruction variant of the intersection in question, the values of initial expenditure and the times of implementation of activities vary. These parameters, for each of the proposed variants, are presented later in the article. In addition, the authors analyze other selection criteria for the optimal option, e.g., social, environmental, security and functionalities.

4.1.1. Description of the Designed Condition—Option 1

Construction of a roundabout on a main road is planned at 30.00 m from the original intersection (Figure 6). Due to the need to make thorough changes to the current road system and a significant scope of works related to this, the partial occupation of land belonging to the contracting authority but also to private owners is expected. That real estate that is private property requires division and

purchase by the contracting authority. The designed solution includes the construction of 5347.00 m$^2$ of roadway with bituminous surface, 2215.00 m$^2$ of pedestrian-bicycle paths with concrete cube surface and 265.00 m$^2$ of parking lot with bituminous surface. The scope of works covered by option 1 covers a total area of 16,556.00 m$^2$. The implementation of the envisaged scope of works does not require demolition and demolition works, because no local construction objects within the area of interest were found based on the site inspection. As part of the implementation of works covered by option 1, additional green areas will be created, including planting of low-foaming vegetation.

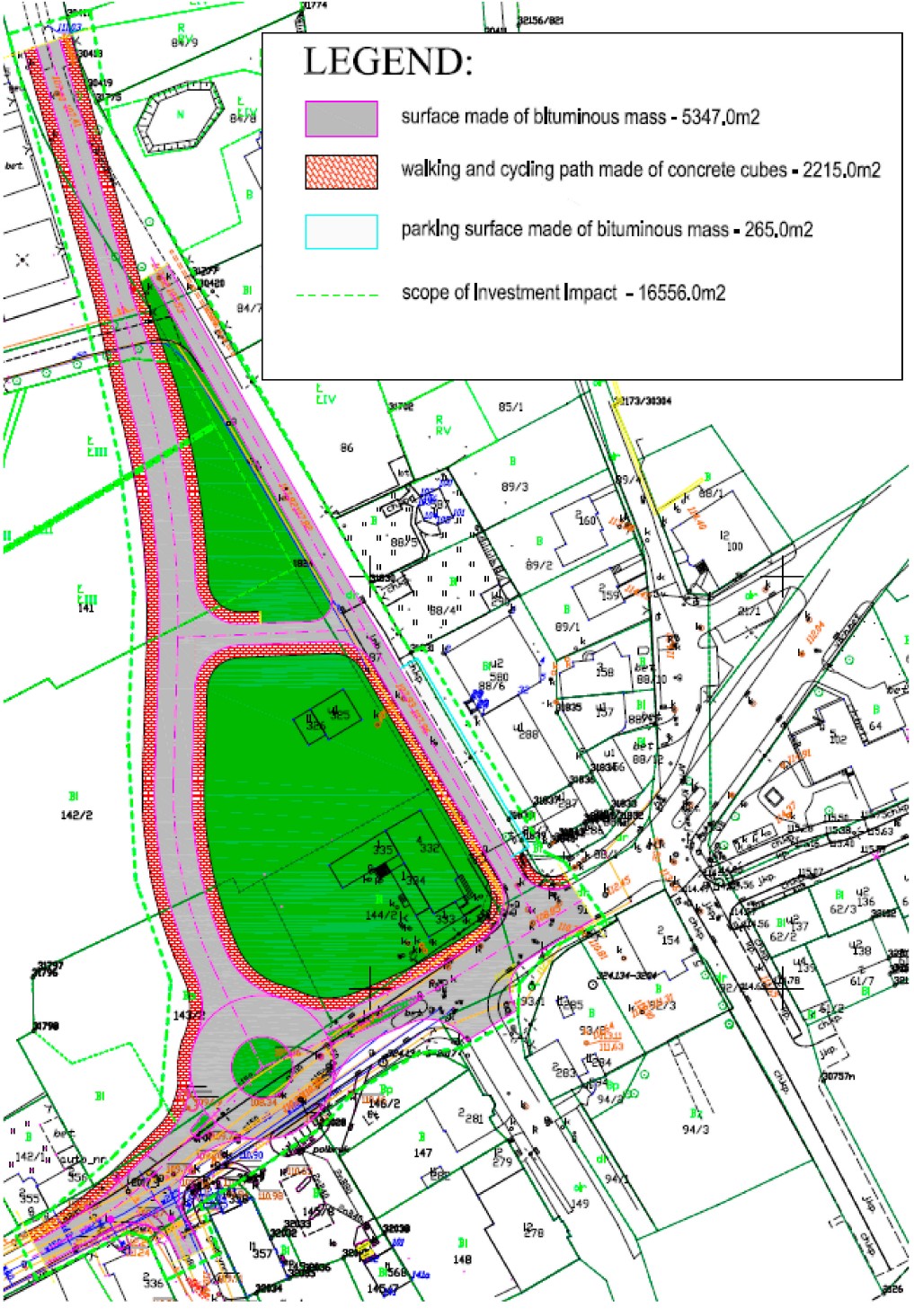

**Figure 6.** Site plan-proposed solution for the road layout at the intersection—option 1.

### 4.1.2. Description of the Designed Condition—Option 2

It is planned to build a roundabout on a main road at the place of the existing intersection (Figure 7). Partial occupation of land belonging to the contracting authority but also to private owners is foreseen. Real estate that is private property requires division and purchase by the contracting authority. The designed solution includes the construction of a road surface with a bituminous surface with an area of 3012.00 m$^2$, pedestrian and bicycle paths with an area of 1512.00 m$^2$ with a surface made of concrete cubes and a parking lot with a bituminous surface with an area of 315.00 m$^2$. The scope of works covered by option 2 covers a total area of 7444.00 m$^2$. To carry out the works, it is necessary to demolish construction works existing within the area covered by option 2.

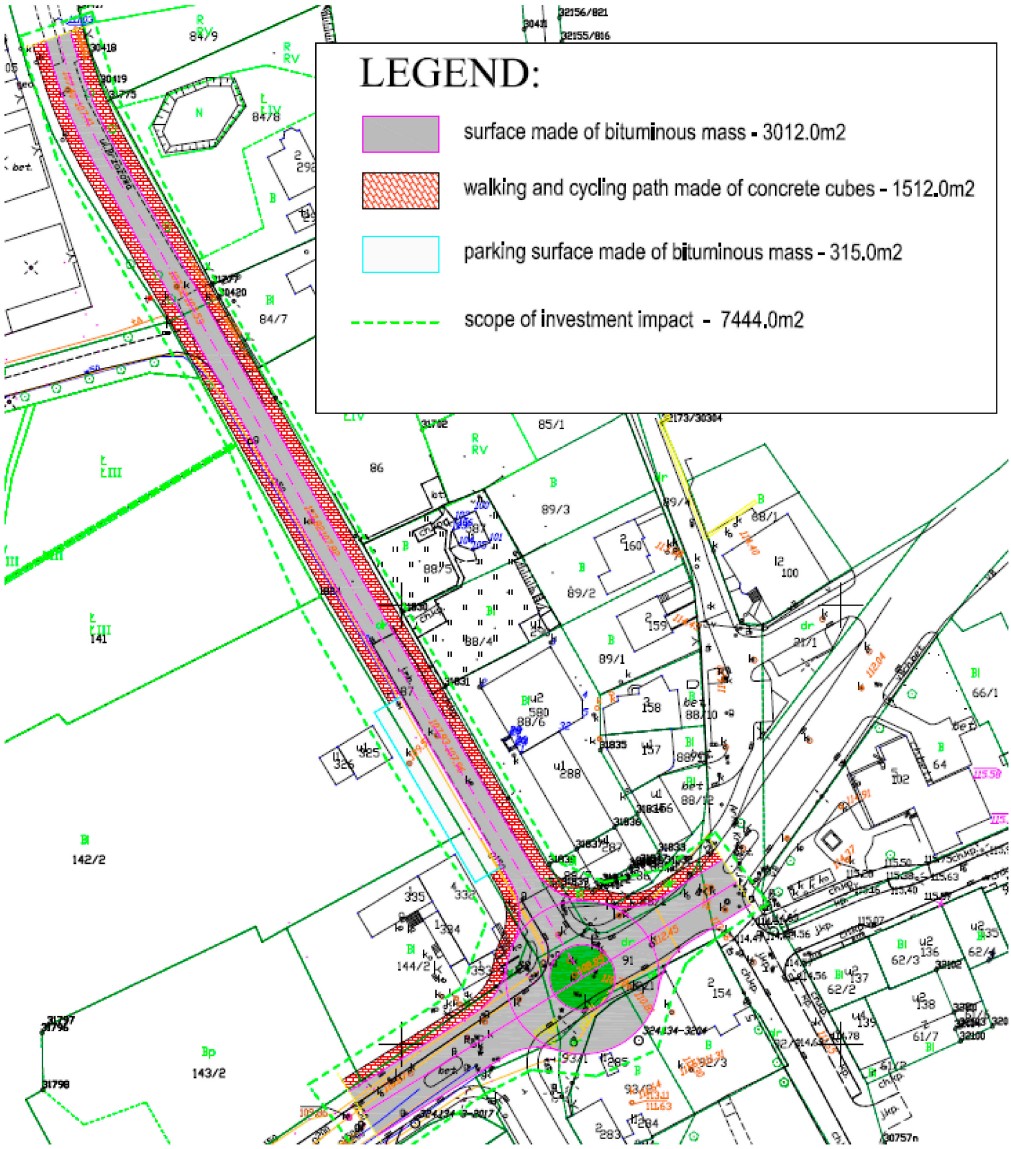

**Figure 7.** Site plan-proposed solution for the road layout at the intersection—option 2.

### 4.1.3. Description of the Designed Condition—Option 3

Construction of a roundabout on a main road at the place of the existing intersection is planned (Figure 8). Partial occupation of land belonging to the contracting authority but also to private owners is foreseen. Real estate that is private property requires division and purchase by the contracting authority. The scope of work includes the construction of a road surface with a bituminous surface

with an area of 2422.00 m$^2$, pedestrian and bicycle paths with a surface of concrete cube with an area of 1740.00 m$^2$ and a parking lot with a bituminous surface with an area of 734.00 m$^2$. The total area on which the works covered by option 3 are designed is 6554.00 m$^2$. To carry out the designed scope, demolition and demolition works of existing buildings should be performed. It is also planned to set the traffic lights in places indicated in Figure 5.

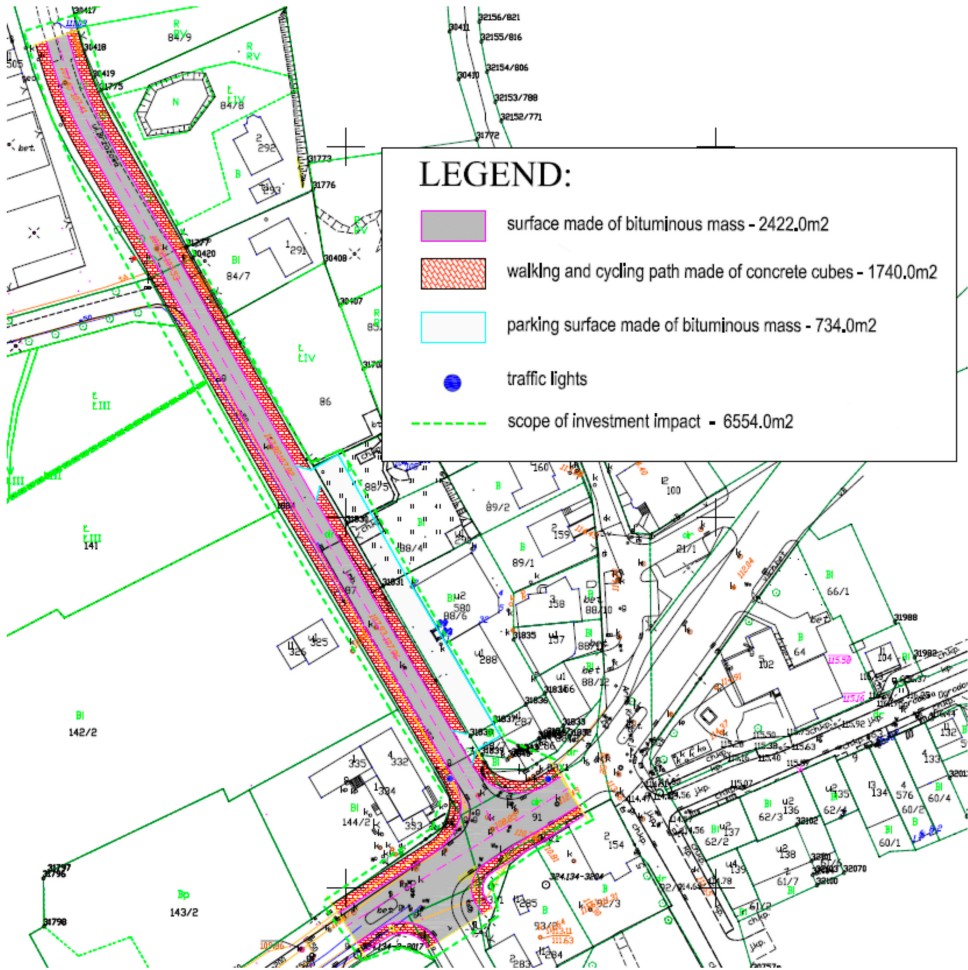

**Figure 8.** Site plan-proposed solution for the road layout at the intersection—option 3.

## 5. Stages of the Optimization Process

A properly carried out optimization process of a construction investment should include the following stages:

1.  Indication of optimization criteria, i.e., determining the parameters that will be analyzed and modeled. In practice, most often these are: the cost of the whole project and its individual stages, times of their implementation, technology and quality of the proposed solutions, size and shape of the structure, scope of investment, standard of finishing the facility, assumed period of its use, costs related to maintenance and use, impact on environment.

2.  Determining the rank of selected optimization criteria, i.e., indicating their degree of importance for the entity performing the process.

3.  Establishment of initial boundary conditions, consisting in indicating actual conditions. These include, among others: location of the investment (e.g., urban conditions), terrain features (shape of the construction plot, soil and water conditions, availability of utilities), scope of the project (conceptual, construction), time limits for the implementation of the investment

(resulting from the use of given technology of works, formal procedures), budget constraints (resulting from the method of financing the investment implementation or current prices on the construction market). Based on the indicated boundary conditions, it is possible to carry out a preliminary optimization process, e.g., costly construction investment. Its important element is, among other things, experience, and historical knowledge, allowing the identification of areas of potential reserves and savings without reducing the quality of works carried out.

4. The proper optimization process, based on the construction cost estimate and investment schedule, considering the initial conditions of the investment. Iterative change of the initial parameters (in specified boundary ranges, while maintaining limitations resulting from regulations, standards, technologies, accepted constructional and material solutions), as a result, indicates the optimal solution. In the case of single-criteria optimization, the result of the process will be unambiguous; for cost optimization it is the total cost of construction, and for time optimization-the date of completion of the project. From a practical point of view, the process becomes significantly more complicated if several criteria are adopted as the basis for deciding. It should also be emphasized that not reducing every parameter, e.g., the cost of performing a given scope of works, will contribute to the reduction of the total investment cost. And so, for example, a change in technology or the use of cheaper material may result in longer works and an increase in indirect costs (related to, among others, maintaining buildings at the construction site). It should then be determined which criterion (criteria) is a priority for the entity performing the optimization.

## 6. Criteria Adopted for Optimization, Selection of the Optimal Variant

The criteria used by the authors to indicate the optimal solution were identified and analyzed during the interview with the contracting authority (the entity that will in practice finance and implement the selected scope of work). The criteria which are rational and justified from the point of view of practical actions that the contracting authority will undertake were analyzed.

Below, the authors present a flow chart of their optimization process, Figure 9.

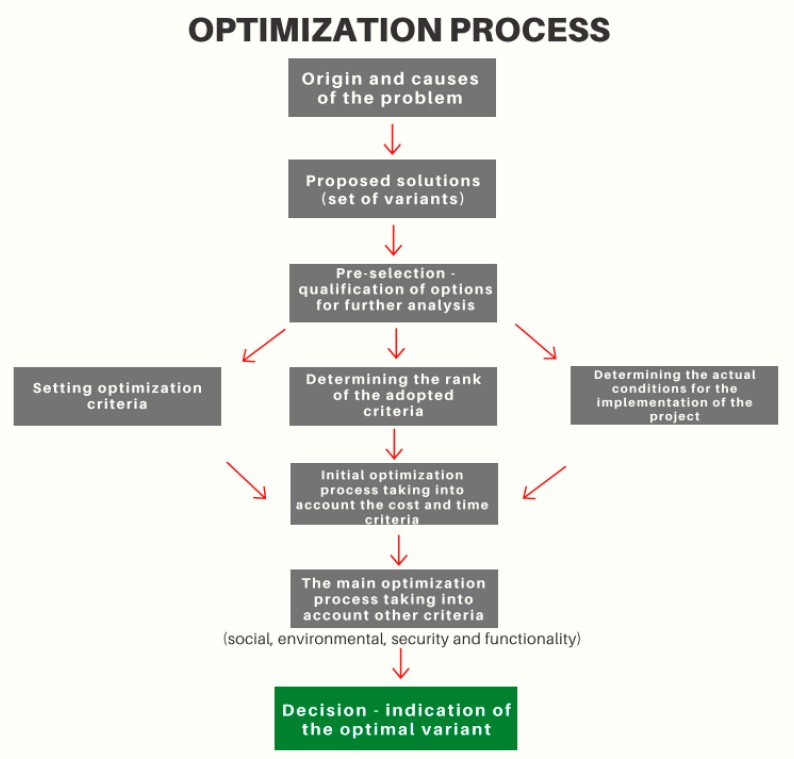

**Figure 9.** Flow chart of optimization process.

### 6.1. Cost Criterion

In order to determine the cost of works, for each of the three proposed variants, project documentation was prepared, on the basis of which a bill of quantities was prepared, in which the scope and amount of works to be performed was determined and assumptions for calculation were adopted. Based on them, three investor cost estimates were prepared.

The following assumptions have been made for the calculation:

- the detailed method used;
- the Sekocenbud system price list has been used, rates and prices from the first quarter of 2020;
- current market prices of materials and information obtained from material producers were also used;
- the following works were taken into account: earthworks, demolition and demolition, foundation layers for the designed pavements, road equipment with road safety devices, land development after completed construction works, road connection with the section that is not subject to reconstruction;
- the cost of the division and purchase of real estate from private owners has been included. Based on the cost estimate prepared, considering the above assumptions, the cost of carrying out the works to be carried out under the three proposed variants was determined. The results are presented in Table 1.

**Table 1.** Statement of cost estimates (excluding value-added tax (VAT)) for the Implementation of the scope of work provided for in options 1, 2, 3.

| Option 1 | Option 2 | Option 3 |
|---|---|---|
| 1,937,994.45 * | 1,590,618.51 * | 1,291,222.97 * |

\* amounts in [monetary units].

Option 1 (Table 1, Figure 6) generates the highest cost of the works planned to be carried out under a given solution (see Figure 10).

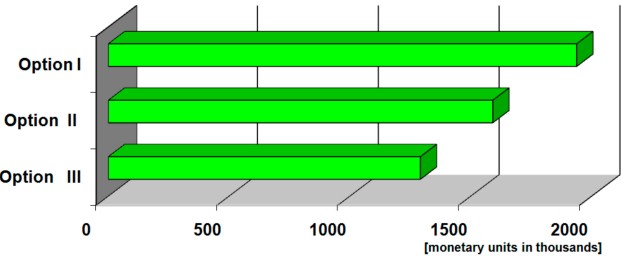

**Figure 10.** Costs of works envisaged for implementation under options 1, 2, 3.

The cost of performing the scope of work covered by option 2 represents 82% of the cost that should be incurred when deciding to implement option 1. The cost of carrying out work according to option 3 represents almost 67% of the cost of performing the scope of work provided for in option 1 (Figure 11).

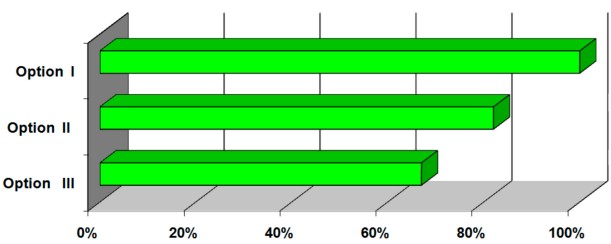

**Figure 11.** Percentages of the costs of works envisaged for implementation under options 1, 2, 3.

### 6.2. Time Criterion

To determine the time of execution of works, construction schedules were prepared for each of the three proposed variants.

The following assumptions were made to the schedules:

- the scope of works results from the project documentation prepared for the three considered variants;
- the works are carried out in the shortest possible time (using the method of uniform work, in some cases parallel and subsequent execution);
- 5 to 10 employees are employed to perform the works (depending on the scope and nature of technological processes);
- the time needed to complete formalities related to the division of land and purchase from private owners should be added to the times given;
- the basis for determining the tangible expenditure of labor and equipment is information obtained from construction cost estimates prepared using a detailed calculation method.

Table 2 presents a summary of the implementation times for the three proposed variants.

**Table 2.** Statement of duration of the works envisaged for the implementation of the scope of work provided for in options 1, 2, 3.

| Option 1 | Option 2 | Option 3 |
|:---:|:---:|:---:|
| 272 * | 165 * | 178 * |

\* times in units [work shifts].

The time related to the implementation of the works covered by option 1 is the longest—272 working shifts (Figure 12). This is primarily due to the largest range of activities to be carried out.

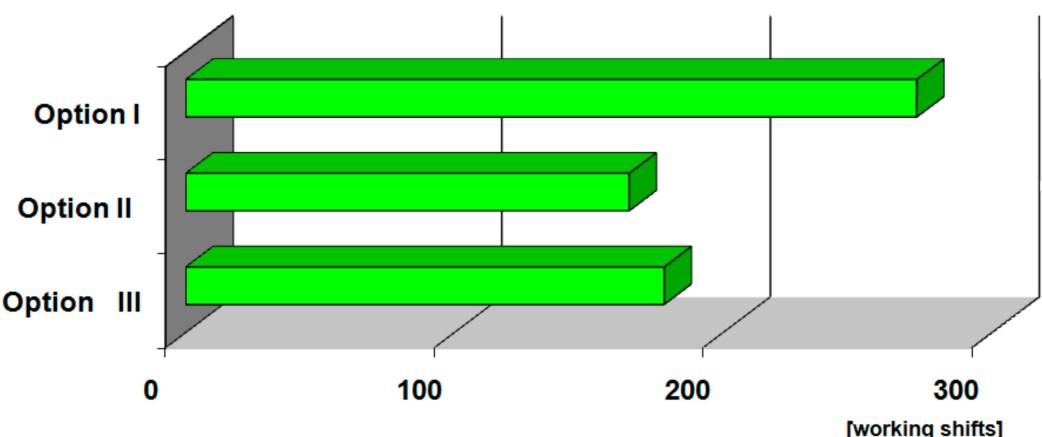

**Figure 12.** Duration of the works envisaged for implementation under options 1, 2, 3.

### 6.3. Social, Environmental, Security and Functional Criteria

To analyze other criteria, which the authors also adopted as the basis for choosing the optimal option, the main advantages, and disadvantages of the three proposed investment options were identified (Tables 3 and 4). The information presented in Tables 3 and 4 should be treated as additional criteria (in addition to the cost and deadline) that are necessary to consider at the investment decision-making stage.

**Table 3.** Advantages of options 1, 2, 3.

| Option 1 | Option 2 | Option 3 |
|---|---|---|
| the slightest interference in the buildings around the road, no need to demolish facilities located in the future investment, the most noticeable effects-there will be a significant flow of vehicles and pedestrians, the highest level of functionality and security of the solution. providing additional safety to municipal road users, by reducing traffic. | similarity of traffic organization to existing. | the least interference in neighboring areas, least logistically troublesome scope of work to be carried out, elimination of potential causes of conflicts with owners of areas adjacent to the road, no need for extensive demolition and demolition works. relatively lowest implementation cost compared to options 1 and 2. |

**Table 4.** Disadvantages of Options 1, 2, 3.

| Option 1 | Option 2 | Option 3 |
|---|---|---|
| the need for major reconstruction of the existing road structure, interference in many private plots (divide them, buy them out and change their qualifications), the long time needed for the division and purchase of plots from private owners. the highest investment cost compared to options 2 and 3. | the need to purchase plots around the planned roundabout from private owners, the need to make numerous demolitions and demolitions of existing-currently used residential buildings, high costs associated with acquiring new areas, the long time needed for the division and purchase of plots from private owners and the demolition and demolition of buildings. | poorly synchronized traffic lights will cause congestion on access roads, preservation of the existing organization of pedestrian and cyclist traffic will cause an additional safety risk due to the increase in road capacity and traffic volume, reconstruction of the existing road system does not solve most of the contracting authority's problems in this area. |

Each of the proposed variants of reconstruction of the intersection generates a specific cost and time of implementation. They are closely related to the scope of planned construction works. At the stage of making the investment decision, i.e., choosing the optimal variant, one should also consider other criteria, important primarily from the point of view of the safety of road participants. When making a decision regarding the choice of a particular investment implementation option, the contracting authority is also obliged to take into account the growing demand for road transport, the ever-increasing requirements of users regarding the quality of travel, an increase in the number of vehicles traveling on the roads, successively increasing traffic volume and sensitivity of road systems to interference and fast infrastructure degradation. An important parameter of choosing the optimal solution should also be the element of transport impact on the environment. This means that the subject of interest should be such a variant, the implementation of which gives a chance to reduce the adverse impact of the investment on the natural environment [32], i.e., it will reduce air pollution and noise emissions in the project impact zone.

Considering the presented analyses and arguments, the authors recommend option 1. This is the most expensive solution among those analyzed and with the longest implementation time, however, taking into account the long-term prognosis of the direction and scope of changes to the existing standards and requirements for road infrastructure, it can be stated that option 1 meets them to the highest degree, and also has the greatest potential [33]. The envisaged solution ensures high standards of the quality of road infrastructure use in terms of functionality, capacity, technical parameters, as well as the safety of traffic participants related to the smoothness of the journey, and reduction of the number of collisions and accidents. As a result of the implementation of the indicated solution

(option 1), it is also expected that the negative impact of the investment on the environment will be significantly reduced.

## 7. Concluding Remarks

The considerations and analysis carried out in the article and analysis of selected variants of road investment implementation justify the formulation of the conclusions and statements presented below:

1.  The implementation of a construction investment includes several complex processes that require rational planning, including in terms of cost and time. Optimization of costs and construction time includes the balance of profits and losses, and as a result the indication of the optimal solution considering the two main parameters of the investment project. In practice, however, it is important that before making the final investment decision, a broad analysis, also considering criteria of a different nature (social, environmental, etc.) is necessary. In some cases, these criteria may significantly influence the awarding entity's decision regarding the choice of investment implementation option.

2.  Due to the range of possible decisions and actions, the process of solution optimization considering various criteria should be started as early as possible, i.e., already at the design stage. This approach allows more possible options and a perspective and comprehensive look at the investment to be obtained. At this stage, optimization can include the size and shape of the structure, type of material and technology of the object or proposed structural solutions. Analyses related to the optimization of solutions, carried out at the design stage, seem rational also from the point of view of the efficiency of spending. It should be emphasized that as the scope and size of investments increase, the complexity of analyses increases, but at the same time the possibilities for optimizing solutions increase.

3.  Due to the complexity of the construction process, in order to properly optimize (taking into account cost, time, functionality, safety, environmental, social criteria), extensive knowledge and extensive experience of the team responsible for carrying out this process (architects, structural and installation designers, engineers, experts in the field of cost analysis, technology and organization of construction works, lawyers, logistics specialists) are necessary. Optimizations of solutions should be carried out many times and at various stages of preparation, sometimes also the implementation of investments, considering current circumstances, but also the direction and scope of forecast changes in the long run. The recommended solution (option 1) takes into account the long-term forecast of road traffic intensity, as well as other parameters ensuring the maintenance of road safety at the highest level, and consequently a reduction in the number of dangerous road incidents and personal, material, environmental and economic losses.

4.  The analyses presented in the article and the proposed solution for the reconstruction and modernization of the intersection (option 1) rationally consider current and future transport needs, expectations of road users and social requirements. In a broader context, this also meets the objectives of infrastructure sustainability, with a focus on its reliability, user safety and limited environmental impact.

5.  It is important to forecast the direction and extent of future changes in relation to the needs of safe and economical transport, expected standards of quality of use (functionality, capacity and technical parameters) and safety of road participants (smooth transition, reduction of collisions and accidents and negative impact on the environment). The presented example and the proposed scope of actions envisaged for implementation under option 1, supported by the analysis of costs and time, can be considered in a broader context as a proposal of actions taking into account the long-term vision of sustainable development of infrastructure in urban and extra-urban areas.

**Author Contributions:** Conceptualization, M.S. and B.G.; data curation, M.S.; formal analysis, M.S. and B.G.; methodology, M.S. and B.G.; writing—original draft, M.S. and B.G.; writing—review and editing, M.S. and B.G. All authors have read and agreed to the published version of the manuscript.

**Funding:** This research received no external funding.

**Conflicts of Interest:** Authors declare no conflict of interest.

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
