# Peer review of "Technical and Economic Analysis of the Implementation of Selected Variants of Road Investment"

_buildings, doi:10.3390/buildings10060097_

Round 1

Reviewer 1 Report

The aim of the paper entitled Technical and Economic Analysis of the Implementation of Selected Variants of the Road Investment is to present the optimal solution from the point of view of the entity responsible for managing a given road section. Overall the abstract of the paper sounds good but the article unfortunately doesn't bring any novelty. The abstract is very similar to introduction section. There is no described results in it. Introduction should be extended. The methodology used in this paper is poor. There are described only typical criteria. Stages of the optimization process should be presented on the flow chart. There is a lack of mathematical basis for the optimal road variant. 

Author Response

Dear Reviewer, our answers are attached below.

Reviewer 2 Report

The article analyzes three different alternatives for modernization and reconstruction of a road intersection to indicate the first best alternative. In the analysis, the authors account for multiple criteria such as cost, time, social and environmental factors that influence the suitability of the proposed project. The authors provide rich information on the need for good road infrastructure to meet the prime requirements of quality, safety and environmental sustainability and the challenges associated with finding an optimal solution. I learned things from the paper about optimization of infrastructure solutions, so in that sense, the article is a contribution to the field that will help to advance the understanding of how to implement efficient infrastructure solutions. However, I have several concerns over the paper that the authors need to address. 

The first concern relates to the disposition of the paper and the ability to convey the paper’s contribution to the literature. Generally, the disposition would benefit from rearranging to make the paper more fluent and the content easier to grasp. For example, the section stating the purpose of the paper, Section 4, should come before the sections describing the optimization process. Actually, this section should come in the paper’s introduction to give it more edge and a better sense of what the paper’s aim and objectives are. The authors should also clarify their research objectives and the contribution they make. Similarly, the description of the research problem and the three alternatives under evaluation should come before the optimization. Otherwise, the paper lacks focus and the optimization sections seem strayed from the paper. Moreover, it makes reading the paper rather tedious. 

The leads to the second concern of the paper, which is the methodology. The authors present the general optimization process, criteria, types of optimization, methods and stages at length. However, when it comes to the actual application of the optimization to the paper’s research problem and how the results are derived, the reader is left in the dark. For instance, it is unclear which of the mentioned optimization methods that is used and why, why they choose the three solutions (the authors mention there are six out of which they look at three, but don’t give arguments for the choice), and specification of parameters etc. is insufficient. Moreover, data, tools and software used (if any?) are poorly presented. Without such methodology, it is difficult to link the optimization to the research problem and the conclusions. I suggest that the authors revise the methodology section and put more emphasis on how the actual optimization is done and how the results are derived. 

Third, the literature review is poorly done. In its present shape, the review of the literature is not thorough enough to give the reader adequate background to the topic or to support the authors’ claims. The authors need to clarify how the setup of infrastructure analysis in the paper differs from the setup in previous literature, and what new results we can learn from this. A more thorough literature review would give more body to the paper and make it more interesting to read. In a nutshell, the whole analysis should be more structured, and linked with current literature both in theory and practice of road infrastructure.

Last, the maps over the proposed solutions are rather blurry and difficult to read. It is hard to make out the details of the maps and hence get a picture of what the proposed solution looks like. They should be sharper and less busy. Also, Tables 3 and 4 with the pros and cons should be improved (less text, align text left instead of center etc.).

All in all, I believe that in spite of these suggestions for more explanation, the paper also could save a lot of space by being shortened and clarified. Aiming for a sizeable reduction in space and more focus on the essentials might be an attractive option for this paper.

Author Response

(The authors gave the same response as above.)

Reviewer 3 Report

  • The abstract should include some conclusions of your paper.
  • Line 160: Define what "poviat" is or delete it
  • Line 162: Use local inspection instead of local vision
  • Line 162: What is the name of this place?
  • Line 190-191: unclear. Please rephrase
  • Line 296: Define VAT
  • The conclusions are too large. Especially conclusion 5.

Author Response

(The authors gave the same response as above.)

Round 2

Reviewer 1 Report

Authors have taken into account all suggestions from the first revision. Paper has been improved and should be published. 

Author Response

Dear reviewer, thank you very much for the favorable reviews

Reviewer 2 Report

The authors made only cosmetic changes to what I thought were major concerns with the paper’s readability and methodology. These cosmetic changes made the paper a bit more transparent and improved the disposition, but as they are merely cosmetic my original concerns remain.

My first concern was that the disposition of the paper is poor and that the paper's aim and objectives are not expressed adequately. The abstract is improved on this point but I maintain my opinion as this is still not clear from reading the introduction and should be more explicitly expressed here, and not in section 3. It should be stated in a simple and straightforward manner from the beginning what the aim and objectives of the research are. Otherwise, the article lacks a focus and appears to have no contribution to research whatsoever. Moreover, the new information at the end of the introduction actually made the paper worse.

The same is true for the authors’ attempts to justify the optimization and make it more transparent. The additions of how the criteria are identified and the flow chart are welcome and improves the methodology slightly but the issue still remains. The methodology is poor and the authors need to be more explicit about how the optimization is done and provide some mathematical representation or similar to support it.

Lastly, adding a couple of extra references does not do the trick to improve the literature review if the authors do not write anything further about them to give the paper more background and body. It is still not sufficient.

Author Response

Dear reviewer, thank you very much for your comments, below is our answer file.

Round 3

Reviewer 2 Report

I find that the new revisions have improved the paper and I am pleased with the authors' efforts and clarifications. The paper is acceptable now.